# Extensive Gaseous Emissions Reduction of Firewood-Fueled Low Power Fireplaces by a Gas Sensor Based Advanced Combustion Airflow Control System and Catalytic Post-Oxidation

**DOI:** 10.3390/s23104679

**Published:** 2023-05-11

**Authors:** Xin Zhang, Binayak Ojha, Hermann Bichlmaier, Ingo Hartmann, Heinz Kohler

**Affiliations:** 1Institute for Sensor and Information Systems (ISIS), Karlsruhe University of Applied Sciences, Moltkestr. 30, D-76133 Karlsruhe, Germany; xin.zhang1023@googlemail.com (X.Z.); binayak.ojha@h-ka.de (B.O.); 2Ulrich Brunner GmbH, Zellhuber Ring 17-18, D-84307 Eggenfelden, Germany; bichlmaier@brunner.de; 3Deutsches Biomasseforschungszentrum Gemeinnützige GmbH (DBFZ), Torgauer Str. 116, D-04347 Leipzig, Germany; ingo.hartmann@dbfz.de

**Keywords:** wood-log firing, combustion air flow control, gas sensors, oxidation catalyst, exhaust quality monitoring

## Abstract

In view of the tremendous emissions of toxic gases and particulate matter (PM) by low-power firewood-fueled fireplaces, there is an urgent need for effective measures to lower emissions to keep this renewable and economical source for private home heating available in the future. For this purpose, an advanced combustion air control system was developed and tested on a commercial fireplace (HKD7, Bunner GmbH, Eggenfelden, Germany), complemented with a commercial oxidation catalyst (EmTechEngineering GmbH, Leipzig, Germany) placed in the post-combustion zone. Combustion air stream control of the wood-log charge combustion was realized by five different control algorithms to describe all situations of combustion properly. These control algorithms are based on the signals of commercial sensors representing catalyst temperature (thermocouple), residual oxygen concentration (LSU 4.9, Bosch GmbH, Gerlingen, Germany) and CO/HC-content in the exhaust (LH-sensor, Lamtec Mess- und Regeltechnik für Feuerungen GmbH & Co. KG, Walldorf (Germany)). The actual flows of the combustion air streams, as calculated for the primary and secondary combustion zone, are adjusted by motor-driven shutters and commercial air mass flow sensors (HFM7, Bosch GmbH, Gerlingen, Germany) in separate feedback control loops. For the first time, the residual CO/HC-content (CO, methane, formaldehyde, etc.) in the flue gas is in-situ monitored with a long-term stable AuPt/YSZ/Pt mixed potential high-temperature gas sensor, which allows continuous estimation of the flue gas quality with an accuracy of about ±10%. This parameter is not only an essential input for advanced combustion air stream control but also provides monitoring of the actual combustion quality and logging of this value over a whole heating period. By many firing experiments in the laboratory and by field tests over four months, it could be demonstrated that with this long-term stable and advanced automated firing system, depression of the gaseous emissions by about 90% related to manually operated fireplaces without catalyst could be achieved. In addition, preliminary investigations at a firing appliance complemented by an electrostatic precipitator yielded PM emission depression between 70% and 90%, depending on the firewood load.

## 1. Introduction

Currently, there is still a heavy worldwide dependence on fossil fuels for heating private households. This is the case also for Germany because there are about 12 million single-room fireplaces and one million central heaters in use, both operated with wood-log fuel for residential heating. In this context, one would expect that the use of biomass, especially wood residuals for residential heating, should be further promoted because heat from biomass combustion is well-accepted as a renewable energy source [1]. However, there are several critical aspects to be considered as well, for example, (i) the high emissions of un- and partly combusted gas components [2,3] and fine particle mass (PM) [4], which contribute considerably to local exhaust emission levels that have been verified to be associated with considerable health risks [5,6], and (ii) the availability of biomass, especially residual wood, for direct combustion purposes is limited. The use of non-residual wood, on the other hand, is in direct competition with other routes of utilization. The advantages of using wood as construction material for building purposes are twofold: First, it contributes considerably to reducing extensive grey energy by substitution of concrete and related cement production. Second, wood is a carbon sink for decades. Moreover, there is no doubt about the historical attractiveness of high-quality wood as a standard material for the construction of furniture. Furthermore, (iii) in terms of newer efforts for the substitution of fossil fuel, the biorefinery of biomass by the production of synthesis gas and subsequent cleaning of the raw syngas for the production of liquid fuel in bio-fermentation processes [7,8] or direct transformation to electric energy in high-temperature solid-state fuel cells (SOFCs) gains more and more relevance. In relation to solid biomass, the direct use of syngas is more flexible, e.g., as fuel for heating, in gas turbines for electricity generation, and also their application in diesel engines has been discussed [9].

Despite those very attractive and promising research activities to transform residual biomass to noble fuels, the use of wood residuals for direct heating in low-power residential combustion appliances enjoys high acceptance and valuation in Europe, especially by the population living in the countryside. Therefore, also in the future, heating homes by combustion of wood will play an essential role because the investments for those wood-log fueled systems are relatively low, and in many regions, the firewood can be made disposable at rather low costs by private engagement. However, depending on the individual landscape and regional density of the households, this kind of residential heating has been identified as one of the major sources of environmental pollution [2,10,11], and in many cities, it has a strong impact on local air quality [12]. Correspondingly, home heating with firewood contributes to a conflict. Governments must take care of people’s health. Therefore, they are requested to amend their emission threshold values to fulfill this goal: e.g., the air quality standards of the European Union, which themselves are requested to be oriented on the latest formulation of the WHO air quality guidelines (2021).

Traditional wood-log fueled low-power combustion appliances, which are operated without automated combustion air mass flow control (CAC) and without any secondary measures of flue gas cleaning like catalytic post-oxidation or particulate matter (PM) removal, represent the dominant wood-log firing technology in private homes. Or in other words, most of the more than 70 million single-room fireplaces in the EU are outdated [13]. Consequently, due to their unequivocal high contributions to air pollution and despite their undisputed high potential as one of the most important sustainable energy sources for private home heating, their political acceptance is a subject of highly controversial discussion. The high level of emissions of those low-power fireplaces is mainly a consequence of the complexity of the combustion process, which in all stages of batch combustion is inhomogeneous and not well reproducible. Due to the fragmented size of the wood-logs, the stacking of the fuel is only less reproducible but takes considerable influence on the combustion quality. The burning of wood pieces step by step from the surface to the bulk fuel is a complicated, non-steady state oxidation process dominated by poor reproducibility of the combustion air flow distribution between the stacked wood-logs and highly influenced by oxygen diffusion processes from the wood-log surface to the inner parts of the wood-piece. This means that wood-log combustion in one combustion chamber is never a complete oxidation process due to more or less dominating unavoidable zones where combustion proceeds under conditions of oxygen deficit. In an early work, Nussbaumer [14] published the correlations between the parameter values which define the main combustion process conditions like combustion temperature (Tc) and residual oxygen concentration (ROC) in the exhaust with the concentration of unburned pollutants. For combustion with improved quality, the two-stage combustion process was recommended, with primary air injection into the fuel bed and secondary air injection into the post-combustion chamber. This allows a good admixing of the combustible gases with air to enable high combustion efficiency and correspondingly low concentrations of pollutants. In the post-combustion zone, where the exhaust gas is post-oxidized by an additional dosage of secondary air, there is a much better, but still not perfect, admixing of the pyrolysis gas with the combustion air related to the situation in the fuel bed. The combustion optimization potential in this zone is given by optimizing the gas retention time and by the sufficiently high reaction temperature. Both parameters play an essential role with respect to combustion kinetics and efficiency. The better the constructive features of the fireplace to enable such high combustion quality, the lower the emissions of toxic exhaust gases and PM can be expected [15,16].

In view of the need for further reduction of gaseous and PM emissions from such manually operated low-power wood-log fueled fireplaces, several attempts have been initiated in the past, such as cleaning of the flue gas [17]. In one of the latest publications, the integration of an oxidation catalyst into the flue gas tube was described, and a small-scale electrostatic precipitator was complemented to reduce the PM-emissions [13]. A clear reduction of the PM_10_-emissions was achieved, but no significant influence of the catalyst was reported. The use of a Pd/Pt-catalyst in the flue gas tube quite far away downstream of the combustion process could be the reason for the insignificant effect of the catalyst [13]. This shows that the combination of low-power wood-log fueled appliances with an oxidation catalyst does not automatically result in a significant decrease in emissions. In addition to being an appropriate choice of catalyst material (morphology, porosity), it needs a well-suited positioning of the catalyst in the hot zone of the fireplace [18,19,20,21] and—as a consequence of the current understanding of the firing process—a continuous and well-defined adjustment of the combustion air streams. The latter can never be satisfactorily done by manual control because even well-trained operators with a good understanding of the firing process are unable to estimate the actual process situation (temperature, required air stream, etc.) by eye. By keeping this aspect in mind, the need for self-adjusting combustion air control systems to ensure optimum excess air in practice by the introduction of sensors for monitoring of Tc, ROC, and CO in the flue gas was already proposed by Nusbaumer [14].

But what would such a CAC concept look like? For CAC of wood-log fueled and wood-pellet fueled central heaters, the monitoring of Tc and ROC in the flue gas is state of the art. However, this concept is based on the assumption that there is a strong correlation among ROC (lambda-value), Tc and combustion-quality pollutant emissions, respectively. This strong correlation is given by the combustion of liquid or gaseous fuels like light fuel oil, gasoil, gasoline, or domestic gas, which all are provided in well-defined, reproducible quality. However, this is no longer true when the wood-logs are burned in batches because fuel composition and combustion conditions change continuously over time and volume. This was experimentally demonstrated by simultaneous measurements of Tc, ROC, and incompletely combusted components like CO, CH_4,_ or formaldehyde [22] over a whole wood-log batch combustion process. Following the basic investigations of Nussbaumer [23], the first automatically controlled wood-log firing experiments were already published in 2009 [24]. The air streams were continuously calculated from the sensor signals of Tc and ROC for estimation of the primary air stream, and from the sensor signals of Tc and the CO/HC-content, the adjustment of the secondary air stream was calculated. During the firing process these values were re-calculated and re-adjusted every 15 s. In this early work, the efficiency of an oxidation catalyst positioned in the post-combustion chamber was already studied. These preliminary results showed clearly that substantial reduction of the gaseous emissions needs (i) an automated control of the combustion air stream well-adjusted on the individual type of combustion appliance and (ii) a suitable oxidation catalyst well defined in material, morphology and size, and well positioned in the hot zone of the fireplace. This means the appliance has to be complemented with motor-driven shutters for air stream adjustment controlled by the signals of air mass flow sensors in separate control loops.

With respect to the measurement of the combustion air flows, the ROC and the combustion temperature, there have been well-established sensors commercially available, with confirmed long-term stabilities under harsh application conditions as part of the automobile power combustion train for many years. However, no stable sensor element for continuous in-situ measurement of the CO/HC-content in hot exhaust gas was available in the past. As already stated above, such a sensor plays a key role in an efficient wood-log combustion process because it is not only essential for proper adjustment of the combustion air streams but also enables continuous monitoring of the quality of the actual gaseous emissions and valuation of the combustion quality in general. This motivated studies of several alternative sensor concepts operated in the flue gas of wood-log firing processes in the last 10 years [25,26], and the studies are still continuing.

In this paper, the technical possibilities for substantial reduction of the emissions in accordance with the considerations addressed above were experimentally studied, and the results are reported. In Section 2, the concept of an advanced, long-term stable CAC system based on the sensor signals for Tc, ROC, and CO/HC-content is introduced, and in Section 3, the experimental settings for evaluation are described. The features of the CAC are illustrated in more detail in Section 4.1, and the levels of emissions reduction achieved by several combustion experiments in context with the complementation of the system with an oxidation catalyst are illustrated in Section 4.3 and Section 4.4. Special emphasis was taken on the performance of the CO/HC-sensor evaluated in the flue gas and periodically checked and regenerated by electrochemical methods (Section 4.4) [27]. Finally, some conclusions and an outlook are presented in Section 5.

## 2. Method of Sensor-Based Air Stream Control for Optimization of the Combustion Process

As already addressed in Section 1, the CAC is based on the separated adjustment of two combustion air flows by repeated measurements of each of two sensor signals: The sensor signals for Tc and ROC are used to control the primary air stream and the sensor signals representing the Tc and the CO/HC-content for setting the secondary air stream. An overview of the experimental setup is given in Figure 1a. The gas sensors are installed in the flue gas tube at a distance of about one meter from the fireplace output (Figure 1b). The Tc is generally measured by a thermocouple inside the post-combustion zone. If a catalyst is implemented, the thermocouple is positioned in contact with the oxidation catalyst plate (Figure 1c) at the side stream downwards of the flue gas. All the sensor signals characterizing the actual combustion situation are measured in time intervals of five seconds, and the corresponding air streams are calculated by several algorithms differently parameterized in different phases of the batch combustion. From manifold experiments, it was deduced that a batch combustion process has to be divided into an ignition phase (IP), a high-temperature phase (HTP), a burn-out phase (BOP), a reloading phase (RLP) and a high-temperature phase after reloading (HTPR) of wood-logs on the fire bed. This phase-specific parametrization of the algorithms was necessary to properly describe all situations of combustion regarding the specific combustion conditions. After calculation, each of the two air streams is adjusted separately by control of the motor-driven shutters in time intervals of 15 s by a separately nested control algorithm based on air mass stream sensor signals (HFM7, Bosch GmbH, Gerlingen, Germany). Further, for continuous operation of the firing process, transition conditions between different phases of combustion had to be well-defined.

Short description of the control strategy: in general, the wood burning kinetics in the combustion chamber where the wood is stacked, i.e., where the heating power mainly develops, is considerably influenced by the combustion air stream (primary air stream, Figure 1a) injected into this zone. The post-combustion in a well-separated reaction zone (Figure 1a) should favorably proceed at excess oxygen concentration conditions and at high enough temperature to enable high pyrolysis gas oxidation kinetics. This means, in all situations of a batch-wise wood-log combustion process, the primary air stream (Figure 1a) has to be optimized this way to enable quick temperature increase in the post-combustion zone. After this ignition phase, a long settling time at a favored temperature range (about 400–600 °C in the case of operation with an oxidation catalyst) is desired. Together with the adjustment of the secondary air stream (Figure 1a), the combustion conditions at the catalyst have to be arranged in this way that there is enough oxygen available for the complete oxidation of the pyrolysis gas components. This means the (relatively cold) secondary air stream has to be adjusted low enough to avoid too much cooling in the post-combustion zone but high enough to enable optimum post-oxidation reaction kinetics. In addition, in case of an oxidation catalyst implemented, for high efficiency of catalytic conversion, the total combustion airstream should be adjusted as low as possible to maximize the contact time of flue gas penetrating the porous catalyst. These considerations indicate that (i) the parametrization of the CAC algorithms depends not only on the individual construction of the fireplace but also on the situation of operation with/without catalyst, and (ii) the parameter adaptation will lose quality with the loss of catalytic activity by aging of the catalyst.

### 2.1. Aspects of Continous In Situ Flue Gas Analysis by Gas Sensors

In this sub-section, a short overview of the high-temperature gas sensors used for continuous flue gas analysis is given.

Monitoring of ROC: For continuous in-situ estimation of the oxygen concentration, a commercial solid electrolyte Pt/YSZ oxygen sensor LSU 4.9 (Bosch GmbH, Gerlingen, Germany) is used. This device combines a Pt/YSZ/Pt Nernst concentration cell (YSZ: Yttrium Stabilized Zirconium Oxide) with a diffusion-limited Pt/YSZ/Pt-coulometric cell [28]. The sensor element is operated at an electrochemical state at which all gas molecules, which reach the internal Pt-electrode via a diffusion channel, equilibrate chemically and thermodynamically with the oxygen (Equation (1)). In case of oxygen access (lambda > 1), the residual oxygen molecules are electrochemically reduced to O^2−^ -ions (Equation (2)) and in an electric field, they are transported out of the coulometric cell from the inner Pt-cathode to the outer Pt-anode via the YSZ solid electrolyte and re-oxidized (Equation (3)).
O_2_ + nHC + mCO ↔ rCO_2_ + sH_2_O,(1)
O_2_ + 4e^−^ → 2O^2−^(2)
2O^2−^ → O_2_ + 4e^−^(3)

The electric current Ip of the ion transport relates to the diffusion-limited current of the oxygen molecules into the reaction cell and, correspondingly, is linearly dependent on the residual oxygen concentration (ROC) in the flue gas. At this point, it may be stressed that the ROC is measured in thermodynamic equilibrium with all other gas components at the operation temperature of the sensor element (about 600 °C) in the presence of the platinum electrodes, which act as an electrode as well as an excellent oxidation catalyst.

The great robustness and stability of the LSU 4.9 oxygen sensor have been confirmed in the exhaust gas of diesel motor cars over the years. Its sensitivity behavior over a wide range of ROC was experimentally investigated in relation to a classical Pt/YSZ/Pt-Nernst concentration cell [29]. A rather good linearity of the sensor signal Ip, especially at oxygen excess situations (λ > 1), was found, which demonstrates the clear advantage of the LSU 4.9 sensor over the classical Nernst cell for combustion air control of wood-log combustion appliances.

Monitoring of CO/HC: In the past, the development of a wood-log combustion air control concept, as introduced above, was restrained because CO/HC-sensor elements with sufficient long-term stability enabling continuous in-situ monitoring of the CO/HC-content in the flue gas were not available. Faced with the relevance of this sensor element for biomass combustion control, in the last five years, several investigations were started to close this gap by further development of an existing sensor device [30] and introduction of new sensor concepts [25,31,32]. At the actual technological status, there is only one type of sensor, which is commercially available and has gained sufficient sensitivity and long-term stability at in-situ operation in the flue gas of wood-log combustion processes. This Au,Pt/YSZ/Pt—sensor chip belongs to the family of high-temperature mixed potential devices. The planar solid-state electrochemical cell consists of high-temperature, stable, ceramic materials like alumina and YSZ, on which a Pt-reference electrode, an AuPt-mixed potential sensing electrode and a heater are microstructured by screen printing. The technical details of fabrication and design are described in [26,33].

Detailed studies of the sensor performance when exposed to ambient air and to model gases like CO admixed with synthetic air over about 100 days revealed clear aging effects, which were mainly observed as sensitivity loss. These instabilities have been indicated to be correlated with changes of the Au-distribution in the AuPt-electrode material with operation time [30] and are likely attributed to the formation of an additional oxide phase at the metallic parts of the triple-phase-contacts [34]. By modification of the AuPt-mixed potential electrode material (more homogeneous AuPt-distribution) and careful adjustment of the operation temperature (600 °C), the long-term stability of this type of gas sensor could be substantially enhanced and, with a deeper understanding of the reasons for sensitivity loss, an electrochemical procedure for sensor regeneration by cathodic polarization, possible at the site of installation, could be developed. Moreover, sensitivity loss due to aging effects was indicated by a clear increase in the electrode impedance [34]. This allows a check of the sensor performance in predefined time intervals and regeneration on demand [27]. Mixed potential sensor elements of this new, modified type (LH-sensors, Lamtec Mess- und Regeltechnik für Feuerungen GmbH & Co. KG, Walldorf (Germany)) are meanwhile available on the sensor market for combustion applications.

When applied for flue gas analysis, the mixed half-cell potential of an AuPt-electrode represents many different gas components that are electrochemically oxidized/reduced [33,35]. This means besides the reduction reaction with oxygen, all oxidizable/reducible components of the exhaust gas more or less contribute to mixed potential formation dependent on their electrochemical reaction kinetics at the AuPt-electrode surface. For exhaust gas analysis of wood-log combustion, this means that an equivalent CO concentration (COe) has to be defined, which relates to the CO-concentration as the leading flue gas component, but also represents the other components like CH_4_, aldehydes, alcohols, and so on, which individually contribute to the mixed potential formation dependent on their individual electrochemical oxidation rate at the AuPt-electrode surface. This functional dependence of the mixed-potential φ_m_ from c(O_2_), c(CO) and other oxidizable HC gas components i with concentration c_i_ was first formulated by Miura [35]:(4)$$\varphi_{m}=\varphi_{o}-a\ln c\left(CO\right)-\sum_{i}b_{i}lnc_{i}(HC)+c\ln c(O_{2})$$

In Equation (4), the constants a, b_i_ and c depend on the reaction rates and φ_0_ is a constant electric potential that relates to the sensor signal U = φ_m_ − φ_ref_ = 0 at synthetic air exposure, whereas φ_ref_ is the potential of the Pt/YSZ-reference half-cell [34].

## 3. Experimental

### 3.1. Setups for the Development of a Sensor-Based Combustion Airstream Control System

In order to evaluate the toxic emissions from wood-log combustion processes at different quality states of CAC development, major wood-log firing experiments have been conducted on an HKD7 single-room fireplace. These experiments were complemented by firing experiments in an SF10SK fireplace to expand the long-term stability studies of the CO/HC sensors. Both firing appliances were provided by Brunner GmbH, Eggenfelden (Germany). Each test-setup was complemented by a gas sensor for measurement of ROC (LSU4.9, Bosch GmbH, Gerlingen, Germany) and by different modifications of mixed potential CO/HC sensors (Lamtec Mess- und Regeltechnik für Feuerungen GmbH & Co. KG, Walldorf (Germany)) to be evaluated. All gas sensors were installed into the flue gas tubes. A representative setting with the HKD7 fireplace is given in Figure 1b. The SF10SK single-room fireplace was operated with CAC software, developed by ourselves, which was already available from earlier studies [26]. In general, after each four wood-log batch firing experiment, the sensors were taken out of the flue gas tube and were exposed to model gases for evaluation of their sensor performance (see Section 3.3).

For further development and adaptation of already existing CAC algorithms [26], the HKD7 fireplace had to be complemented with some further devices. The chimney draught, Tc and the combustion air stream were measured by a differential pressure sensor (DMU/A-U/V1, FuehlerSysteme eNET International GmbH, Nürnberg (Germany)), a thermocouple (K type), and two air mass flow sensors (one HFM7 (Bosch GmbH, Gerlingen, Germany) for each air stream channel (Figure 1f)), respectively. The combustion airstreams were not only measured but also controlled by motor-driven shutters installed at the inlet of the air stream channels (Figure 1d). They were adjusted to desired flows as calculated by the control algorithm based on Tc, ROC and CO/HC emissions as introduced in Section 2 and described in more detail in Butschbach et al. (2009) [24], Kohler et al. (2018) [22], and Ojha et al. (2017) [26]. For continuous referencing of the IR-active components of the exhaust, an FTIR gas analysis instrument (Gasmet, Ansyco GmbH, Karlsruhe, Germany) was used. The instrument enables continuous extraction of the hot exhaust (5 L/min), which is led into the heated FTIR measurement chamber for IR-analysis via a heated tube and particle filter. Thus, analysis of hot flue gas in time intervals of 10 s is possible. This instrument played an important role in the development and optimization of the CAC algorithms and experimental adaptation of the software parameters on the individual combustion appliance.

It has to be noted that the HKD7 fireplace was taken from the production series but modified to enable independent settings of the primary and secondary air stream. This fireplace was additionally equipped with an oxidation catalyst (EmTechEngineering GmbH, Leipzig, Germany) (Figure 1c). The catalyst in the shape of a plate is a porous Al_2_O_3_ foam ceramics (300 × 300 × 18 mm^3^, porosity 10 ppi) coated with a metal oxide/(Pt, Pd (10g/ft^3^))-layer.

For field tests, the same setup of the HKD7 fireplace as described above was installed in the pilot plant for wood-log fueled furnace systems (Brunner GmbH, Eggenfelden, Germany), but some operational conditions were different. First, the chimney draught was nevermore adjusted by an exhaust gas ventilator but was continuously measured by the differential pressure sensor. In this way, the finalized automated CAC system could be evaluated under natural chimney draught conditions. Second, the analysis of the gaseous emissions was limited to continuous measurements of the CO-concentration (URAS 14, ABB Group, Zürich, Switzerland). CO is the major gas component of incompletely combusted gas emissions. Finally, unlike the setup in the ISIS-laboratory, this test setup was complemented by an electrostatic precipitator (Öko Tube Inside, OekoSolve AG, Mels, Switzerland). This allowed comparative experiments under different (manual/automated) combustion air control, with/without catalyst, and optional electrostatic particle precipitation. The PM-measurements were done according to a standard gravimetric procedure using a Wöhler SM96 sampling system (Wöhler GmbH, Bad Wünnenberg, Germany). Particle sampling started three minutes after reloading the wood-logs on the fire bed and was conducted for 30 min.

### 3.2. Stacking of the Wood-Logs—Combustion Air Guiding

To get the firing experiments as reproducible as possible, the stacking of the wood-logs in a reproducible manner is a prerequisite. This still does not guarantee optimal reproducibility of the firing experiments because wood-logs, even split from the same kind of wood, are not well defined in shape, thickness and even density, and the firing conditions, like the temperature of the fireplace body and the chimney draught take influence on the combustion performance. Therefore, exceptionally pre-dried beech wood-logs (residual water content lower than 20%) were stacked in a rather fixed geometrical arrangement (Figure 2), if not especially indicated. Ignition was started on the top of the wood stack (upside-down combustion) using a commercial igniter.

The combustion air streams are guided from a wide slit above the front door top down along the inner side of the front door window to the wood-log stack (primary air) and by the horizontal slits at the side walls (Figure 2) upwards into the combustion chamber (secondary air). To obtain a more complete understanding of the gas streams inside the combustion chamber, see also Figure 1c.

### 3.3. CO/HC Gas Sensor Sensitivity Checks and Calibration

Two exemplars from each of the three Au,Pt/YSZ/Pt high temperature mixed potential gas sensor types (Section 2) differing in Au/Pt-ratio of the electrode material, namely CS10K, LH64, and LH68 from the production series of Lamtec Mess- und Regeltechnik für Feuerungen GmbH & Co. KG, Walldorf (Germany), were investigated. As already mentioned above, typically, after each of the four combustion experiments (each lasting about four hours), all six sensors were taken out of the flue gas tube for sensitivity tests at exposure to model gases (1000 ppm CO/5% O_2_ and at 100 ppm H_2_/5% O_2_). These tests were conducted in the laboratory by use of an automated gas mixing station [33]. For proof of the sensor stabilities, 24 batch firing experiments were conducted with the SF10SK fireplace, i.e., in total, the sensors were operated for about 96 h in the flue gas. After an interim check of the long-term stability, two individuals of the six sensor elements, which promised the best stability in the exhaust, the LH68-1 and LH68-2 sensor elements, were further operated in the field tests. This means the long-term stability studies of these exemplars relate to a total operation time in the flue gas of about 472 h.

For estimation of the COe from the mixed potential φ_m_ of an LH-sensor element, a calibration route with model gases was defined. Following Equation (4), measurement of φ_m_ at different O_2_/CO/N_2_ enables the determination of the constants a and c. Further, having in mind that ROC is measured by the LSU 4.9 lambda-probe, after calibration of a and c, φ_m_ is determined only by c(CO) if all the other gas components HC with concentration c_i_ are neglected. Of course, this is only an approximation and does not describe the situation in exhaust where several other reactive components take influence on φ_m_. However, fortunately, there is a rather good correlation between the concentration of the different HC gas components in the exhaust gas with CO concentration which allows the calibration of the function COe = f(φ_m_), where COe is the equivalent CO-concentration, which represents approximately all uncombusted components in the flue gas. COe is used as the input parameter for secondary air combustion control and may be a useful parameter for monitoring the quality of combustion in general.

### 3.4. Development of a Combustion Airstream Control Algorithm and Its Validation under Real-Life Operation Conditions

As already stated in Section 2.1, the development of an efficient CAC algorithm adaptable for the automation of different kinds of single-room fireplaces has been extensively studied in the last 15 years [22,24,26]. However, except for [24], most recent studies were carried out on fireplaces operated without the implementation of a catalyst for the treatment of the flue gas. In Section 2, it was already presumed that the control parameter values of the air flow algorithms as fixed at operation without an oxidation catalyst would have to be changed after the complementation of the appliance with such a catalyst. As already mentioned above, the fireplace used in this work for experimental development of the advanced CAC algorithms in the ISIS-laboratories, HKD7-1# (Brunner GmbH, Eggenfelden, Germany), was equipped with a new oxidation catalyst (CAT1#, Figure 1c), but in a first run, it was operated with the airstream control algorithms as developed in earlier work with the identical fireplace but without catalyst. These experiments were done to compare the results with those achieved after adaptation of the control parameter values on the firing system complemented with the catalyst to evaluate (i) the influence of the catalyst itself on the emission quality and (ii) the adaptation of the control software on the specific combustion conditions necessary to achieve optimized catalytic flue gas post-combustion.

For validation of the whole system under real-life operation conditions, the experimental setup was reproduced at the pilot plant station of the Brunner GmbH by operation of the same type of fireplace (HKD7-2#) with the same modifications, oxidation catalyst (CAT2#) of the same type as CAT1# and the identical sensors (LH68-1, LH68-2). However, CAT2# implemented in HKD7-2# had already been operated in this fireplace in different manually operated batch firing procedures. Overall, CAT2# had already been used for the combustion of about 2000 kg of beech firewood. Adaptation of the air stream control algorithms as developed for HKD7-1#/CAT1# to firing experiments with HKD7-2#/CAT2# did not fail but revealed clearly higher emissions as observed with the identical algorithms from the experiments with HKD7-1#/CAT1#. Moreover, even after considerable modification of the control parameter values, the emission results were clearly worse comparable to those achieved when the original algorithm was run with HKD7-1#/CAT1# (compare Figure 3 with Figure 4). Whereas, after the substitution of CAT2# by a new CAT3# and re-adaptation of the control parameters, finally, the combustion air control characteristics were found to be more similar to that with HKD7-1#/CAT1# (compare Figure 5 with Figure 3), except for the behavior in the ignition phase. These differences in the emission results achieved with the different catalysts give valuable evidence of the influence of the oxidation catalyst on the combustion quality. This will be discussed in Section 4.

**Figure 3 sensors-23-04679-f003:**
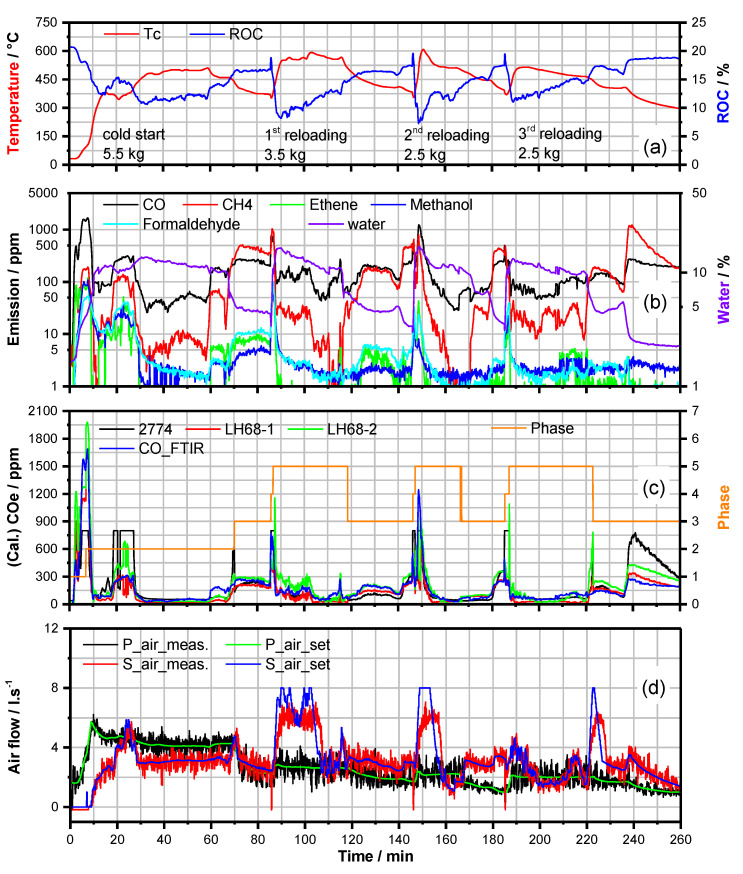
Batch combustion process values of a firing experiment with HKD7-1#/CAT1# after finalization of the process control software. Ignition with 5.5 kg, reloaded with 3.5/2.5 kg beech wood-logs. (**a**) Course of the combustion temperature Tc and the ROC. (**b**) FTIR analysis of the exhaust over time. (**c**) Course of the FTIR CO-analysis in comparison to the COe-analysis calculated from three mixed potential gas sensor signals LH68−1, LH68−2 and CS10K (2774). Different phases of the batch combustion process (IP(1), HTP(2), BOP(3), RLP(4), HTPR(5)) are indicated (orange line). (**d**) Combustion air flows vs. time set by the software; green—primary air (P_air_set), blue—secondary air (S_air_set) and measured values, black—P_air_meas and red—S_air_meas.

## 4. Results and Discussion

### 4.1. Adaptation and Optimization of the Control Algorithms for an HKD7 Fireplace with Catalyst

As already mentioned in Section 2, for proper description, the wood-log fueled batch firing process had to be divided into different phases, which are controlled by individual software and parametrization. The IP is characterized by the highest emission rates because, in the first minutes after ignition, the temperature at the catalyst is still too low to gain enough reaction kinetics for post-combustion. Therefore, this phase has to be overcome as quick as possible. In order to gain a quick increase of the combustion temperature Tc, the primary air stream is linearly increased with Tc. Having reached the onset conversion temperature of the catalyst (typically at about 150 °C), the transition to the HTP is initiated. In this second phase, combustion air stream control is characterized by an additional opening of the secondary air stream shutter with increasing Tc and increasing CO/HC concentration and a decrease of the primary air stream with increasing Tc and decreasing ROC (Figure 3).

Approaching the end of the HTP, i.e., beyond the maximum Tc, typically, Tc decreases monotonically, and ROC increases simultaneously because the kinetics of combustion is reduced successively. In a batch firing process, the amount of hot solid residuals (glue and solid residuals/charcoal) related to the amount of not yet burned wood increases continuously, and this leads to a change in the fuel. This altered situation of charge combustion is now defined by the BOP, which is again treated by an individual parametrization of the control algorithm. The transition from HTP to BOP was defined to be initiated when Tc < 0.85 Tmax and ROC > 13.5% are valid (Tmax: maximum temperature in HTP). For example, this transition is observed (Figure 3) at 70 min (phase 2 → phase 3, Figure 3c). In BOP, the primary air stream is decreased with decreasing Tc and increasing ROC. The secondary air stream is predominantly decreased with Tc as well but moderately increased with an increase of CO/HC-emissions.

At a well-defined range TRmin < Tc < TRmax, reloading of firewood is recommended. This is a status of BOP at which still a moderate amount of hot solid residuals is available for proper re-ignition of some further wood-logs reloaded. Reloading firewood on top of these hot residuals can generate a completely different situation of the combustion process related to the first ignition of wood stacked in a cold combustion chamber (Figure 2a). Now the re-ignition of reloaded firewood is characterized by a highly non-steady state situation of combustion because the ignition is started in the hot chamber from the bottom of the wood stack deposited on top of the hot residuals, which, perhaps, provides a very high amount of heat. Too much glue means that re-ignition (outgassing of the wood-logs) would be too quick: therefore, there would be a sharply increasing oxygen consumption (sharp decrease of ROC to very low values). This can lead to uncontrollable combustion situations, which means a quick transition to combustion at oxygen deficit even at fully opened air shutters because not enough combustion air can be provided. On the other hand, too low an amount of hot residuals leads to delayed re-ignition, or ignition may even fail. In this case, partial smoldering is observed at rather low Tc leading to incomplete combustion correlated with high amounts of toxic gas emissions.

To control the combustion air flows even in this highly non-steady state situation, the RLP had to be developed. After reloading (door is closed again), the RLP algorithm starts with a continuous increase of the primary air stream in steps from the last value adjusted before reloading. Simultaneously, a sharp increase of secondary air stream is allowed up to about 6 L/s but again decreased with increasing ROC. Of course, after ignition, Tc may also rise sharply. At Tc beyond a defined temperature (here 435 °C), the transition to HTPR is initiated. The corresponding control software is quite similar to that of the HTP but with some modified parameter values due to the differences in combustion situation: that is, wood-logs are now put on hot residuals and burning from the bottom. Tmax is defined again for HTPR, and the conditions for transition to BOP are the same as defined in HTP.

The final status of the parametrization of the control software was tested by standard beech wood-log stacking (ignition) and different distributions of the wood-logs after reloading on the hot residuals (Figure 2). Figure 3 represents the emissions of a typical combustion experiment conducted from ignition at room temperature to burn out after three reloading sequences. It is well demonstrated that Tc rises to values above 150 °C already within about eight minutes after ignition (transition to HTP (phase 2)) and does not exceed 600 °C. In general, ROC decreases when Tc increases due to increasing combustion kinetics with Tc. At this final stage of development of the CAC software for firing experiments in the laboratory, only two very short peaks of c(CO) > 1000 ppm are observed, one at IP and one at second reloading. At most of the time intervals of the complete charge combustion process, c(CO) is registered clearly below 400 ppm, and in many time intervals, even below 100 ppm. Obviously, the critical combustion situations directly after reloading are also well covered by the control software, which is demonstrated by a quick decrease in the emissions and a subsequent sharp drop to rather low values in terms of about two minutes (Figure 3).

### 4.2. Validation of the Control Algorithm under Real-Life Operation Conditions

After adaptation and optimization of the CAC algorithms at HKD7-1#/CAT1# in the laboratory at ISIS, control software and an experimental setup with an identically constructed fireplace (HKD7-2#) were operated under real-life conditions in the pilot plant of the Brunner GmbH over four months. As already mentioned in Section 3, the experimental conditions were partly different (natural chimney draught, only CO-analysis of the flue gas available). At ignition situation, the draught of the cold chimney was typically quite low (3–5 Pa, depending on the weather situation) and increased to usual values (10–20 Pa) with continuing wood-log combustion process.

At first, parametrization of the CAC software needed significant modification for adaptation on the HKD7-2# fireplace when operated with CAT2# (Section 3.4). As illustrated in Figure 4, even after several iterations of experimental adaptation of the parameter values, obviously, the markedly low emissions of CO and unburned HC components, as reported with CAT1# (Figure 3), were not achieved. At IP and after each reloading sequence in most cases, for short time intervals, the CO-emissions were clearly higher than about 2000 ppm (Figure 4), and the minimum CO-emissions in the HTP were not lower than 500 ppm and after reloading rarely lower than 100 ppm. This was interpreted by a lower catalytic oxidation activity of CAT2# due to aging effects. Indeed, after the substitution of CAT2# by a new catalyst (CAT3#) of the same type and geometry, more or less the same or even better combustion quality was observed (Figure 5) as achieved with HKD7-1#/CAT1# in the laboratory (Figure 3). These results give rise to the conclusion that both the proper adaptation of CAC parameters on the operation conditions (with/without catalyst) and the activity of the catalyst are the major aspects that essentially determine the quality of combustion and, thus, the flue gas quality.

**Figure 4 sensors-23-04679-f004:**
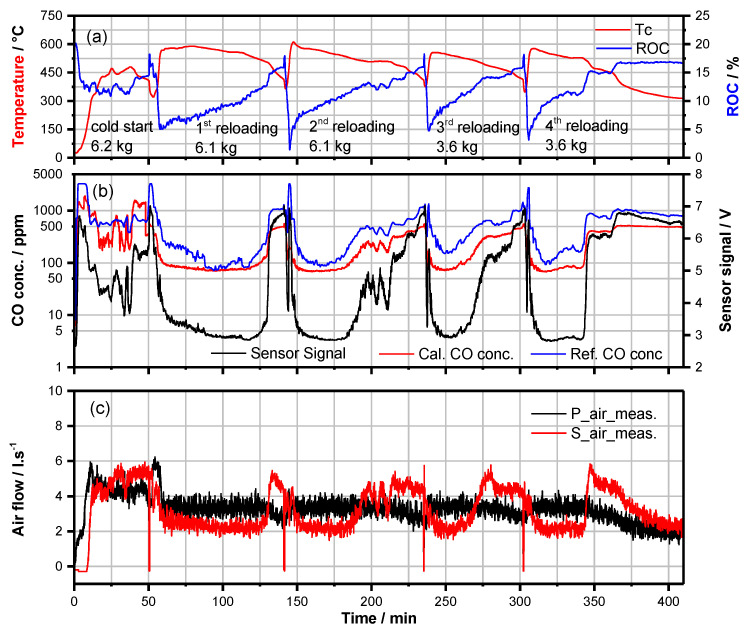
Exhaust analysis data (**a**,**b**) and control data (**c**) vs. time of a batch firing process conducted in a modified HKD7-2#/Cat2# fireplace fueled with beech wood-logs after adaptation of the CAC parameter values (data illustrated as experiment 8 in Figure 7 and as experiment 11 in Figure 9).

In Figure 5a, Tc and ROC vs. time and their dependency on reloading actions are illustrated similarly to those achieved in the laboratory (Figure 3), but now under the conditions of natural chimney draught. Also, the course of the combustion air flows (Figure 5c) is similar as visualized in Figure 3 but significantly lower than those in Figure 4c. The course of COe, as calculated from the signal of LH68-2 (Section 3.3), in some time periods coincides quite well with the CO-concentrations measured with the analysis instrument (URAS 14, ABB group, Zürich, Switzerland), but in others, it differs clearly. The reasons are interpreted to be twofold. First, the lower limit of detection of the LH-sensors for CO was checked to be at about 70 ± 10 ppm. This means in time periods when the CO- and HC-concentrations are considerably lower, this is not measured by the LH68-2 sensor element. Second, COe determination is based on calibration with CO. Therefore, it is expected to differ significantly from CO-analysis in time periods when the contribution of HC- components to φ_m_ are significantly lower/higher than considered in the COe-calibration (Section 3.3). These results obviously disclose the limits of CO/HC-analysis with LH-type mixed potential gas sensors. However, despite these constraints in sensitivity and selectivity, the LH-sensor elements have confirmed excellent robustness and long-term stability (Section 4.3) when operated in the flue gas, and their well-suited sensing characteristics allow valuation of flue gas quality at least at COe-concentrations above 100 ppm. This enables considerable progress in advanced CAC algorithms, as the emission data (Figure 3 and Figure 5) impressively demonstrate.

**Figure 5 sensors-23-04679-f005:**
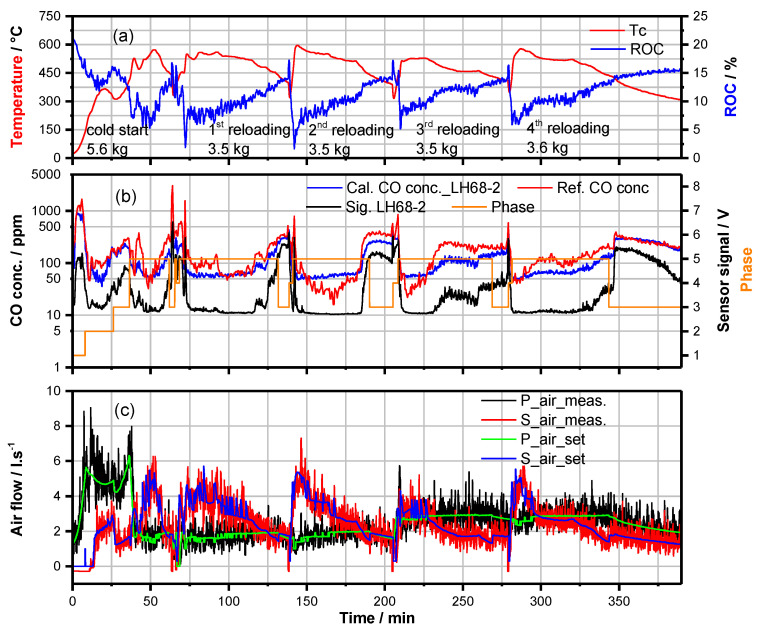
Data of a typical batch firing process of beech wood-logs in HKD7-2#/CAT3# after finalization of the process software. (**a**) Combustion temperature Tc and ROC vs. time. (**b**) The course of the LH68−2 sensor signal (black), COe-value as calculated from the LH68−2 sensor signal (blue) and CO-concentration as referenced by URAS 14 gas analysis instrument (red). The different phases (IP(1), HTP(2), BOP(3), RLP(4), HTPR(5)) of the batch firing process are indicated by the orange line. (**c**) Combustion air streams as indicated in Figure 3.

For a quantitative comparison of the emissions at different operation conditions of the HKD7-1# fireplace, the emission data of whole batch firing experiments from IP (cold start, white) over HTP (bright brown) to BOP (bright blue) were compared (Figure 6) for wood-log charges (about 5.5 kg beech wood) without reloading. First, the emissions without catalyst but with automated CAC, second with CAT1# but operated with the CAC software used without catalyst and third with CAT1# and specific parametrization of the CAC software for operation with catalyst are visualized in Figure 6 together with the course of Tc and ROC (upper diagrams) and of the primary and secondary combustion air flows (lower diagrams).

**Figure 6 sensors-23-04679-f006:**
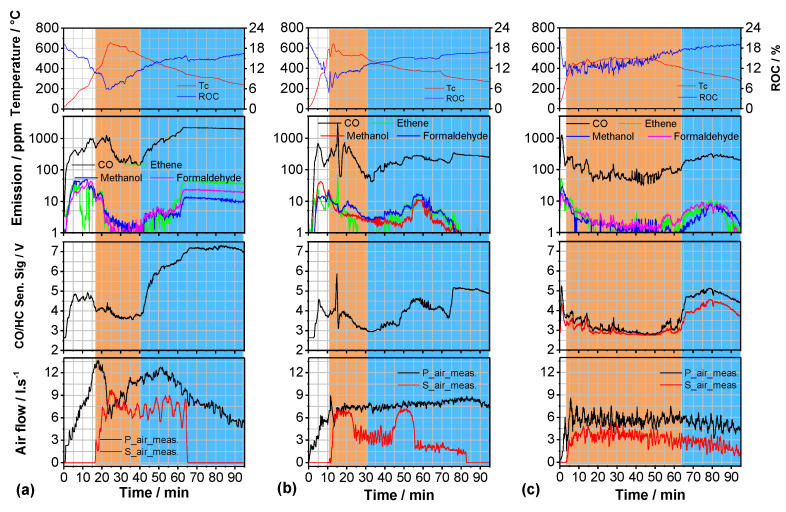
Comparison of the typical combustion air flows and emission characteristics of the HKD7-1# fireplace vs. time of wood-log batch combustion when operated at different conditions. All data are illustrated as indicated in Figure 3. (**a**) CAC as developed for operation without catalyst. (**b**) With CAC–software as developed for case (**a**) but now with CAT1#. (**c**) With CAT1# and specific adaptation of the CAC–software. Tc in (**a**) is measured at about the center of the hemispherical post-combustion chamber (Figure 1), whereas Tc in (**b**,**c**) is recorded in touch with the catalyst at the side gas stream downwards (Figure 1). The white, orange, and blue background represents IP, HTP and BOP respectively.

Without a catalyst (Figure 6a), the transition temperatures (IP → HTP, HTP → BOP) were, of course, set to higher values than in case of operation with a catalyst (Figure 6c) and, correspondingly, the opening of the secondary air shutter was started late at about 380 °C. Implementation of CAT1# without adaptation of the control parameters does not lead to much better combustion quality (Figure 6b). However, after adaptation of the control parameter values, the transition IP → HTP, i.e., catalytic oxidation with CAT1# by additional secondary air stream dosage started already at about 200 °C (Figure 6c), and the transition temperature HTP → BOP was set to a combustion status at which ROC exceeds 13%, and Tc falls below 0.85 Tmax. At all combustion conditions investigated, the beginning of the HTP is well indicated by a relatively sharp drop of the ROC to values less than 12% (Figure 6). However, as Figure 6c clearly illustrates, operation with CAT1# leads to a much shorter IP, and much longer HTP and the CO-emissions are measured to be much lower in concentration compared with the other two operation conditions.

In addition, over the whole batch combustion experiment (Figure 6c), the combustion air flows are adjusted to lower values compared to those as visualized in Figure 6a,b. This is an essential aspect because at the final state of development of the CAC algorithms, the control parameter values were optimized to minimize the (cool) air flows to attain a long contact time of the flue gas with the CAT1#, which enables extended oxidation at a rather high temperature but at enough combustion air to keep ROC high enough for efficient oxidation in the oxygen excess regime.

A more quantitative comparison of the emissions achieved by these firing experiments (Figure 6) is given in Table 1. By multiplication of the sum of the primary and secondary combustion air streams in l/s with c(CO) in mg/m^3^ as measured with the FTIR gas analysis system, the actual relative CO-emission in mg/min was calculated and averaged over the specific phases of batch combustion as indicated.

In Table 1, some key-values as measured/calculated at four different conditions of wood-log combustion are documented. The first column represents some data from a firing experiment already published in [25,26], which was conducted at manual control of a Varia R fireplace (Spartherm GmbH, Melle, Germany) following the instructions of the producer. These data are taken as the “100%-emission reference”. All the other experiments were conducted in an HKD7 fireplace (Brunner GmbH, Eggenfelden, Germany) at different combustion conditions, as already discussed (Figure 6), and their emission data are related to those of the manually controlled experiment. Of course, only a very rough relation of the emissions to manual control operation can be expected from those data because they are based on fireplaces that are similar in size but different in construction. These data are by no means suitable to compare the quality of different fireplaces but may be convenient to gain a first general idea about the level of emission drop to be achieved by automated CAC and additional complementation of HKD7 with CAT1#. A comprehensive comparison of the emissions achieved by hand-adjusted and automated CAC of the Varia R was already discussed in [25,26].

As already mentioned above, the duration of HTP is increased by a factor of three and the average air flow is reduced to 58% when HKD7 is operated with CAT1# (last column, Table 1). The total CO-emissions in HTP are reduced to 43% when automated combustion air control is introduced and to 20% when in addition CAT1# is complemented. Further decrease of the CO emissions to 15% is achieved by specific adaptation of the control software parameter values to operation with CAT1#. Even more impressive are these measures with respect to emission abatement in the BOP. Implementation of CAT1# results in a lowering of CO-emission depression to only 17%, as expected. This value drops further to 6% after the adaptation of the control parameter values. The other emission values show similar trends.

Taking the whole batch combustion emission values into account (without the IP, because there only Tc-conducted primary air stream increase is possible), a CO-emission (mg/m^3^) decreases to 18% without and to 9% with the specific adaptation of the control parameter values to CAT1# achieved. This means the combination of automated CAC, specifically adapted to the type of fireplace and to the specific combustion conditions with a highly active oxidation catalyst, enables considerable shortening of IP and abatement of the CO emissions by more than 90%.

A more comprehensive overview of the results achieved with different oxidation catalysts is given in Figure 7, where the average CO-emissions (mg/min) calculated over the whole charge combustion process of different firing experiments are visualized. All experiments were started with ignition at room temperature (cold start). The data confirm the substantial influence of a well-working catalyst on emission abatement when operated with well-adapted CAC parameter values. Implementation of CAT1# enables a decrease of CO-emissions by about 90%, but only when automated CAC is well adapted (compare experiments 1, 2 and 3, Figure 7). For comparison, the experimental results 3–5 achieved with CAT1# confirm good reproducibility of very low emissions and agree very well with the experiments 10–13 conducted in the field tests with CAT3#. The residual variations of the values of different experiments are mainly related to different reloading sequences and limited reproducibility of stacking due to variations of wood-log shapes.

**Figure 7 sensors-23-04679-f007:**
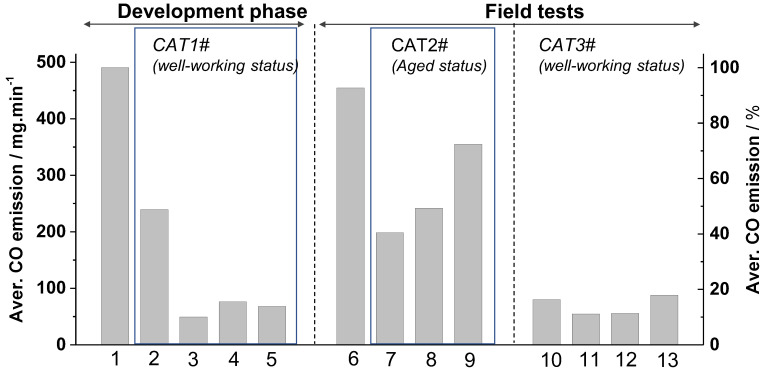
CO-Emissions averaged over the whole batch firing experiments, including IP. The bars (1–13) indicate individual firing experiments at specific conditions. 1–3: Average CO-emissions (HKD7−1#) calculated from the data presented in Figure 6a–c: 1—CAC without catalyst (Figure 6a), 2—CAC with catalyst (Figure 6b), 3—adapted CAC with catalyst (Figure 6c). 4–5: Analogous conditions to 3, but with firewood reloading (compare, e.g., with Figure 3). 6: Experiments conducted without catalyst by manually adjusted air streams (HKD7−2#) but guided by observation of the actual CO-emissions. 7: Manually adjusted air streams like experiment 6, but with HKD7−2#/CAT2#. 8–9: CAC with HKD7−2#/CAT2#. 10–13: CAC of HKD7−2#/CAT3# but at various reloading (different number and weight of wood-logs). Exemplarily, the data of experiment 11 are illustrated vs. time in Figure 5. Relative Aver. CO-emissions (%) are related to the absolute Aver. CO emission of experiment 1.

An interesting exception can be found in experiments 6–9 (Figure 7). Experiment 6 was conducted by manual control of the combustion air streams without a catalyst but guided by observation of the actual CO-emissions (manual continuous emission optimization). This was repeated in experiment 7, but now after the implementation of CAT2#. Experiments 8–9 were conducted with CAC after adaptation of the control parameter values on HKD7-2#/CAT2#. All these emissions achieved by experiments 7–9 (Figure 7) are clearly higher than those observed after proper adaptation of the control parameters values on a highly active oxidation catalyst (experiments 3–5 and 10–13).

These data strikingly illustrate again that low emissions of batch-wise wood-log firing are possible by CAC but only achieved by specific adaptation of the CAC parameters on a highly active oxidation catalyst well positioned inside the firing system. In addition, the data extracted from the CO/HC-sensor open options to monitor and document emission quality over one or more heating periods and signal aging of the catalyst if the emissions increase significantly in case all other conditions like the quality of the firewood are kept constant.

Beyond these very promising results achieved by this advanced CAC system when combined with a highly reactive oxidation catalyst, there was still one question unanswered: Does combustion of different kinds of wood (e.g., beech, spruce, fir, or pine) need a different valuation of the CAC parameters to attain a similar high quality of combustion? First, additional firing experiments in the laboratory with pine wood-logs of about the same volume as usual stacked with beech were conducted with the same setup as described in Section 2 and with the control parameters as fixed by the experiments with beech firewood on the system HKD7-1#/CAT1#. The emission results were significantly higher (100 ppm ≤ c(CO) ≤ 500 ppm) than those achieved with beech wood, and this was correlated with not optimal combustion air dosage and too low a maximum Tc attained. The reasons for these differences were roughly explained by the lower energy density of pine and the higher natural resin content in softwood.

Indeed, a firing experiment (Figure 8) conducted with about the same volume of pine wood-logs (Ignition: 3.7 kg; reload: 2.3 kg) and at general changes of the primary air streams enhanced by a factor of 1.5 and the secondary air streams decreased by a factor of 0.5 yielded CO-emissions at a similar low level (40 ppm ≤ c(CO) ≤ 500 ppm) as achieved with beech-wood (Figure 3). These results clearly demonstrate that adaptation of the control parameters on the kind of firewood is quite simple but essential to attain high-quality combustion behavior.

### 4.3. Comparison of Particulate Matter (PM) Emissions at Different Operation Conditions

As already mentioned in Section 3.1, various field test experiments were conducted under different combustion conditions (manual/CAC, with/without catalyst) and at optional complementation of the fireplace with an electrostatic precipitator. An overview of the PM-emissions resulting from those experiments is given in Figure 9.

**Figure 9 sensors-23-04679-f009:**
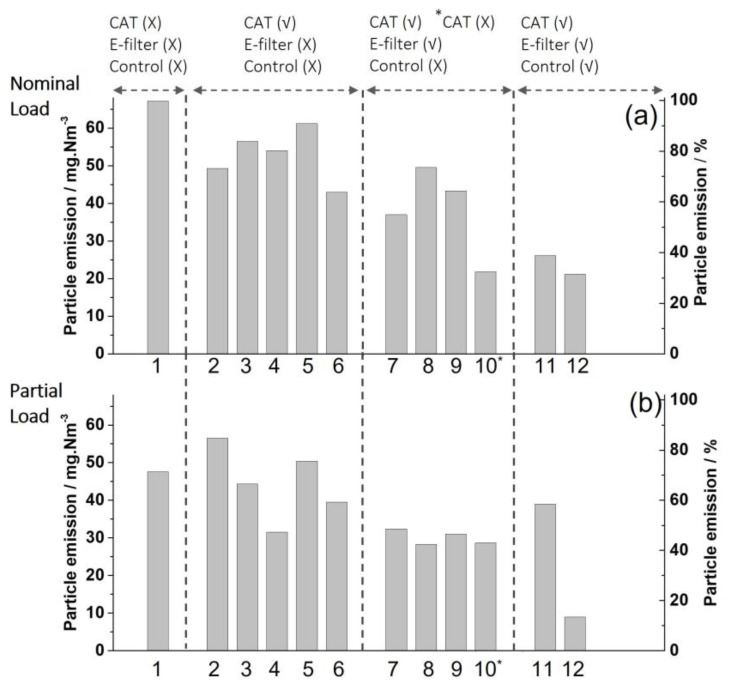
Overview of PM emissions based on various firing experiments with beech wood-logs at different operation conditions. Values were determined gravimetrically over a period of 30 min, started 3 min after reloading, respectively. Each value represents an average over the reloading periods of a firing experiment. Nominal load-ignition/reload: 6 kg/6 kg, partial load-ignition/reload: 6 kg/3.5 kg. Experiments 1, 10*: without CAT, experiments 2–9, 11 conducted with CAT2#, experiment 12 conducted with CAT3#. All other operating conditions are as indicated. Relative particle emissions (%) are related to the emissions of experiment 1 at nominal load (100%).

As indicated by repeated experiments (2–6, 7–9, conditions of operation as indicated in Figure 9) conducted at equal operation conditions, respectively, the reproducibility of the results is not very good. However, a clear trend can be reported. PM emissions measured by experiments without catalyst, electrostatic precipitator (EP) and automated CAC (experiment 1, nominal load) are the highest. These emissions at a nominal load of beech wood-logs are clearly higher than the upper limit allowed by law in Germany (40 mg/m^3^, 1. BImSchV). After complementation of the HKD7-2# with CAT2# (experiments 2–6) but still, without EP, the PM emissions at nominal load are significantly decreased by about 15–20% with noticeable scatter of the individual results. At partial load, the PM emissions are generally slightly lower, but the measurements scatter too much to indicate a clear trend. However, when the EP is additionally operated (experiments 7–10), a remarkable PM emission abatement is indicated, especially at partial load (Figure 9). At nominal load, the scattering of the results is still noticeable. This gives rise to the assumption that at higher PM emissions, the PM cleaning effect by the EP may be less efficient.

The clearly lowest PM-emissions at nominal load (about 20 mg/Nm^3^, Nm^3^: standard volume unit (0 °C, 1.013 hPa)) as well as at partial load (<10 mg/Nm^3^) are achieved at automated CAC of HKD/-2#/CAT3# (experiment 12, Figure 9) with the EP activated. This experiment 12 (Figure 9) is represented in Figure 7 as experiment 13, i.e., it is simultaneously characterized by one of the lowest CO-emissions. At nominal load, PM emissions were decreased by about 70% and at partial load by about 85%. These results may give preliminary evidence that automated CAC, when combined with a highly active oxidation catalyst, allows combustion at lower air flows and, therefore, better efficiency of EP at simultaneously very low toxic gas emissions.

### 4.4. Long-Term Stability of the LH-Sensors

As already stated above, an essential component for advanced automation of the combustion air supply of low-power wood-log fueled fireplaces is the performance and reliability of the sensors operated for exhaust analysis. In this respect, with standard thermocouples for the measurement of Tc and the LSU 4.9 probe for monitoring of ROC, very robust sensor elements are available. However, as already discussed in Section 2.1, a CO/HC-sensor for in-situ application in the hot flue gas with reliable performance and long-term stability was not available up to now. The performance of different modifications of sensors based on the Au,Pt/YSZ mixed potential gas sensing concept was investigated in parallel to the experiments for the development of the CAC algorithms as described above in Section 3.3.

Finally, overall the two most promising exemplars, Au,Pt/YSZ-type II, were operated in the flue gas of about 120 firing experiments and from time to time, the signal base line and sensitivity to model gases (Section 3.3) were checked. The results of these investigations with respect to signal long-term stability are summarized in Figure 10.

At excellent base line stability, the long-term sensitivity of the normalized signal in a reliability corridor of ±20% at hot flue gas exposure over 472 h is illustrated. This quite good stability was further improved to ±10% by application of the cathodic regeneration procedure (Section 2.1). For comparison, in the past, the very best performance of an Au,Pt/YSZ-type CO/HC sensor element (CS10K) showed a long-term sensitivity loss of about 30% within 21 firing experiments (about 84 h of operation in the exhaust) [26].

## 5. Summary, Conclusions, and Outlook

Wood-log combustion experiments in a slightly modified, commercial low-power fireplace (HKD7, Brunner GmbH, Eggenfelden, Germany; NP: 9 kW) complemented by a commercial oxidation catalyst (EmTechEngineering GmbH, Leipzig, Germany) implemented in the post-combustion zone and by an electrostatic precipitator (Öko Tube Inside, OekoSolve AG, Mels, Switzerland), were investigated with the focus to develop an advanced, long-term stable combustion air control system to optimize combustion and minimize emissions in all phases of a batch-wise combustion process. The automation of the batch firing process is based on separate control of primary and secondary air flow determined by combustion air stream control algorithms. These algorithms use the monitoring of the temperature of the catalyst and the in-situ exhaust analysis (residual oxygen concentration (ROC) (LSU 4.9, Bosch GmbH, Gerlingen, Germany) and content of un- and partly oxidized gas components (CO/HC)) for the setting of the combustion air flows in time-intervals of 15 s. Air stream adjustment is achieved by motor driven air stream shutters adjusted in a feedback control loop with a combustion air mass flow sensor (HFM7, Bosch GmbH, Gerlingen, Germany) and is therefore widely independent of chimney draught changes.

The long-term stability of the combustion air flow control system was evaluated by field tests over four months of almost daily firing on the working days. Special emphasis was placed on the signal interpretation and evaluation of the long-term stability of a high temperature mixed potential Au,Pt/YSZ/Pt gas sensor element (LH68-type, Lamtec Mess- und Regeltechnik für Feuerungen GmbH & Co. KG, Walldorf (Germany) for continuous in-situ analysis of the CO/HC-content in the flue gas. Repeated sensitivity and signal stability tests in model gases from time to time while operation in the exhaust over a total of 472 h (about 120 firing experiments) revealed no systematic sensitivity loss or drift effects and confirmed the very good stability of this type of sensor. This already very good sensor performance could be further enhanced by a recently discovered method for sensitivity check (Electrochemical Impedance Spectroscopy) and an electrochemical sensing regeneration procedure to be applied on demand in an environmental atmosphere in situations of firing breaks.

With respect to the long-term stability of the whole combustion air stream control system, the missing link of high-temperature sensor elements suitable for flue gas analysis under harsh conditions is now closed by this CO/HC sensor because the stability of the other sensors used for monitoring of ROC (LSU 4.9, Bosch GmbH, Gerlingen, Germany) or air mass stream monitoring (HFM7, Bosch GmbH, Gerlingen, Germany) is already well proven by their worldwide applications in combustion engines.

The field tests demonstrated that both, a high oxidation activity of the catalyst and the proper adaptation of the combustion air flow control parameters, are important requirements for excellent oxidation efficiency of the flue gas components, i.e., minimization of the emissions. Repeated firing experiments with different amounts of beech wood-log loadings revealed CO-emissions of less than 300 ppm (about 375 mg/m^3^) during 92% and less than 150 ppm (about 188 mg/m^3^) during 35% of the time of a whole batch combustion experiment over 240 min. A rough quantification of these CO-emissions in relation to the emissions of manually controlled firing without a catalyst was estimated to be an emission reduction of about 90%. Comparable experiments with pine wood-logs revealed similar high combustion quality. However, these experiments have shown that simple, common software adaptation of the combustion air streams on the kind of wood is necessary.

Preliminary studies of the particulate matter (PM) emissions revealed only a small decrease of PM by the introduction of the catalyst but a rough decrease by about 40% when an electrostatic precipitator is complemented at manual control of the air streams. Interestingly, the highest abatement of PM-emissions by 70% to 85% (dependent on the amount of firewood loaded) is observed at automated control with the control parameters well adapted on a highly reactive, well-conditioned oxidation catalyst and complemented with a suitable electrostatic precipitator. This remarkable result may be explained by the assumption that the PM emission is significantly depressed when the combustion air streams are controlled rather low by combustion air stream control parameters well adapted to the combustion appliance, which enables efficient post-combustion by a highly reactive oxidation catalyst. This, in turn, may favor the high efficiency of electrostatic precipitation.

The disposability of this new long-term stable Au, Pt/YSZ/Pt mixed potential CO/HC gas sensor and its utilization in advanced combustion air control systems is only one important aspect of realizing low-power wood-log firing appliances which promise emissions much lower than those currently available in the market. In addition, this type of sensor also enables continuous logging and real-time visualization of the exhaust emission quality. In the future, this aspect will gain enhanced importance because the operator of the combustion appliance will get immediate feedback about the actual quality of the batch-wise combustion, and in addition, CO/HC content simultaneously may be logged as a firing quality-protocol over the whole heating season. Furthermore, if wood-logs of high and constant quality (e.g., dryness, log size) are combusted, registration of a significant decrease of the flue gas quality may give evidence of combustion malfunction like decreased reaction activity of the catalyst due to aging effects or any dysfunction of the fireplace itself. This information could be displayed/alarmed at the operator monitor already at an early stage.

## Figures and Tables

**Figure 1 sensors-23-04679-f001:**
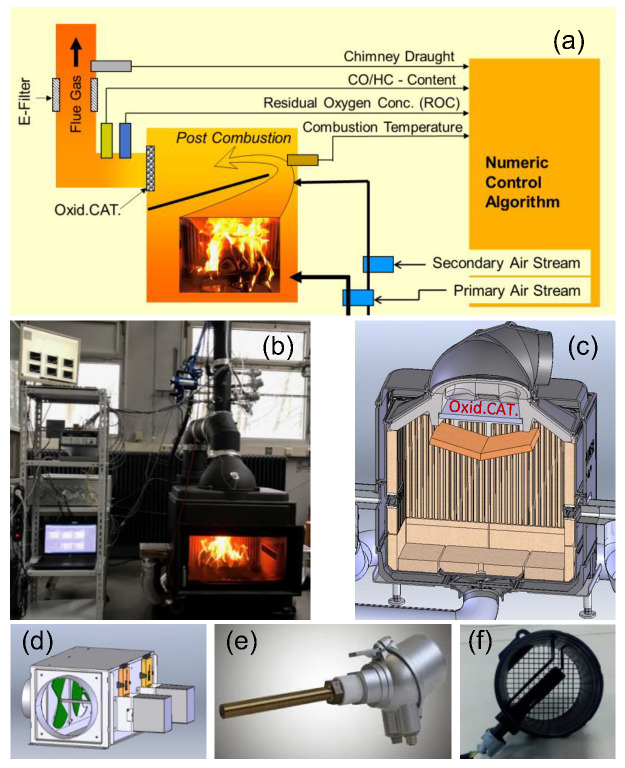
Setup of the firing experiments. (**a**) Schematic overview of the experimental arrangement. (**b**) Arrangement of the firing experiments conducted at ISIS laboratory with the gas sensors and the dark blue/silver box for gas extraction for FTIR analysis placed at the vertical flue gas tube. (**c**) Schematic drawing of the firing chamber (view through the front door) with the slits for secondary air injection at both sides. Flue gas deflection plates are indicated in orange color. The porous oxidation catalyst is represented by a grey plate located between the deflection plates and the hemispherical flue gas tube. (**d**) Step motor-driven shutter system for primary and secondary air flow adjustment. At the front and rear sides, the box is connected with a commercial particle filter (Bosch GmbH, Gerlingen, Germany) as dimensioned for motor truck vehicles. (**e**) Housed CO/HC mixed potential high-temperature gas sensor with local hardware for heating and signal pre-processing. (**f**) Air mass flow sensor (HFM7, Bosch GmbH, Gerlingen, Germany).

**Figure 2 sensors-23-04679-f002:**
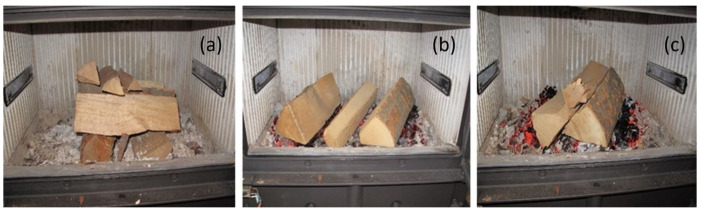
Stacking of the wood-logs. (**a**) Before ignition (5.5 kg beech) and (**b**,**c**) at different reloading (distribution of 3.5 kg beech) on the glue (view through the open front door). Ignition was always started by igniter stripes positioned under the smaller wood pieces of the top layer (**a**).

**Figure 8 sensors-23-04679-f008:**
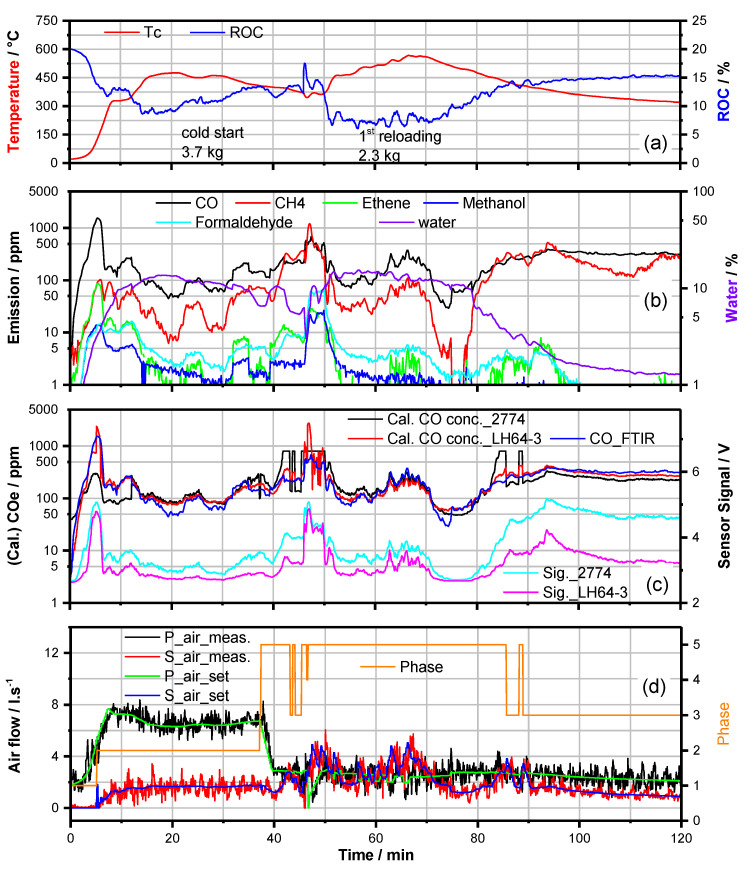
Data of a typical batch firing process of pine wood in HKD7-1#/CAT1#. (**a**) Combustion temperature Tc and ROC vs. time. (**b**) FTIR analysis of the exhaust components, incl. water content over time. (**c**) Course of the FTIR CO-analysis in comparison to the COe-analysis as calculated from two different mixed potential gas sensor (LH64−3, CS10K (2774)) signals. (**d**) Combustion air flows vs. time. Values set by software; primary air (P_air_set), secondary air (S_air_set) and measured values, P_air_meas and S_air_meas. Different phases of the batch firing process (IP(1), HTP(2), BOP(3), RLP(4), HTPR(5)) are indicated (orange line).

**Figure 10 sensors-23-04679-f010:**
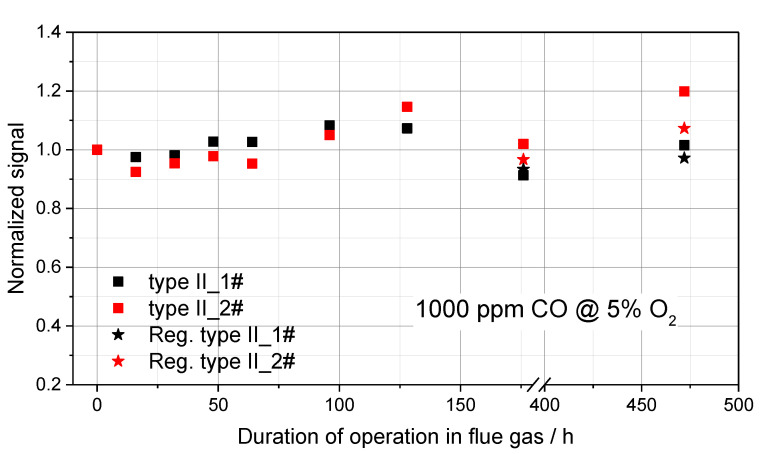
Normalized signal of type II_1# (LH68-1) and type II_2# (LH68-2), Au,Pt/YSZ mixed potential sensor elements as sampled in 1000 ppmCO/5% O_2_ model gas from time to time during operation in the exhaust of wood-log combustion experiments. The values indicated by ★ represent the signals achieved after regeneration by cathodic polarization (Section 2.1).

**Table 1 sensors-23-04679-t001:** Quantitative comparison of the emissions investigated at different conditions of beech wood-log batch combustions as visualized in Figure 6. WFP: Whole firing process.

		Manu. Control	Auto. Control		Auto. Control		Auto. Control with	
		without	without Catalyst		with Catalyst		Catalyst + Improved	
		Catalyst					Algorithm	
		(Varia R)	(HKD7)		(HKD7, CAT1#)		(HKD7, CAT1#)	
	Unit	Value	Value		Value		Value	
		abs.	abs.	rel. (%)	abs.	rel. (%)	abs.	rel. (%)
Tmax	(°C)	405	654	-	642	-	507	-
80% Tmax	(°C)	324	523	-	514	-	406	-
Duration of HTP	(min)	29.7	19.2	-	22.1	115%	57.6	299%
Av.-flow of air stream (HTP)	(L·min^−1^)	302	878	-	588	67%	509	58%
Av.-CO conc. Rel. 13% O_2_ (HTP)	(mg·m^−3^)	1870	300	16%	229	12%	119	6%
Total CO emission (HTP)	(mg)	17,950	7700	43%	3534	20%	2706	15%
Av.-CO emission (HTP)	(mg·min^−1^)	605	400	66%	160	26%	47	8%
Av.-Ethene emission (HTP)	(mg·min^−1^)	13.1	1.4	11%	1.7	13%	0.8	6%
Av.-Methanol emission (HTP)	(mg·min^−1^)	6.9	3.3	48%	2.5	36%	0.99	14%
Av.-Methanal emission (HTP)	(mg·min^−1^)	12.4	1.8	15%	3.6	29%	1.5	12%
Av.-CO conc. Rel. 13% O_2_ (BOP)	(mg·m^−3^)	5005	2596	52%	361	7%	945	19%
Total CO emission (BOP)	(mg)	38,950	38,280	98%	6600	17%	2268	6%
Av.-CO emission (BOP)	(mg·min^−1^)	950	957	101%	120	13%	84	9%
Av.-Ethene emission (BOP)	(mg·min^−1^)	13.4	12.8	96%	2.2	16%	1.3	10%
Av.-Methanol emission (BOP)	(mg·min^−1^)	7.2	5.5	76%	2	36%	1.3	18%
Av.-Methanal emission (BOP)	(mg·min^−1^)	9.9	8.6	87%	3.3	38%	1.7	17%
Av.-CO conc. Rel. 13% O_2_ (WFP)	(mg·m^−3^)	2247	1658	74%	342	15%	456	20%
Total CO emission (HTP + BOP)	(mg)	56,900	45,980	81%	10,134	18%	4974	9%

## Data Availability

Dataset for Figure 3, Figure 4, Figure 5, Figure 6 and Figure 8 is available at https://doi.org/10.5281/zenodo.7919391.

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
