# Peer review of "Extensive Gaseous Emissions Reduction of Firewood-Fueled Low Power Fireplaces by a Gas Sensor Based Advanced Combustion Airflow Control System and Catalytic Post-Oxidation"

_sensors, 2023, doi:10.3390/s23104679_

Round 1
Reviewer 1 Report
The content of the work is important and valuable. The presented technical solutions and research results may have an impact on reducing the harmful environmental impact of commonly used methods of heating single-family houses. However, I have a few comments on the work:
1. In the abstract of the paper, the purpose of the work should be clearly defined. The purpose of the paper should also be further explained in the article at the end of the introduction. The purpose of the work was clearly written down only at the end of the work, in the summary.
2. I believe that the article should describe in more detail the exhaust gas analyzers used in the research (FTIR analyzer, particulate analyzer). The method and place of collecting exhaust gas samples and their transport to the analyzers should be clearly presented.
3. For me, the following expressions are unclear and incomprehensible: ,, application of the wood-logs on glue”, ,,amount of glue” ,,Reloading of firewood on the glue”, ,, of heat provided by the glue”, ,,Too much glue”, ,,too low amount of glue leads”, ,,that wood-logs are now put on glue”, ,, after reloading on the glue”.
4. On page 9, it says that preliminary tests for checking and calibrating CO/HC gas sensors were performed for six sensors. Two were selected for further testing. What were the differences in the six sensors selected for testing. What caused two of them to perform better than the others.
5. In my opinion, there should be a list of abbreviations and designations in the work. The work is extensive. There are a lot of abbreviations and designations in the work. A list of abbreviations and designations placed in one place would make the content of the work easier to read and understand.
6. Sentence from lines 466-467: „The relative amount of glue increases continuously and this leads to a change of the fuel.” is incomprehensible.
7. Sentence from lines 470-471: „This transition is for instance observed (Figure 3) at min. 70.” should be written more clearly.
8. I do not understand the approach to the research, the results of which are presented in table 1. The HKD7 fireplace is compared with the Varia R fireplace. The HKD7 fireplace was automatically controlled according to various algorithms and worked with various equipment. The Varia R fireplace was manually controlled. The operating parameters of the HKD7 fireplace are compared with the operating parameters of the Varia R fireplace. These are two different constructions. Such a comparison is unjustified. The HKD7 automatically controlled fireplace should be compared with the HKD7 manually operated fireplace. In my opinion, only then would the comparison make more sense. It would be justified. The purpose of the work is not to compare different designs of fireplaces (constructions of different manufacturers). It is not to show that the fireplace of one manufacturer is better than the fireplace of another manufacturer, in the absence of compliance with the conditions of comparison. This is a significant shortcoming of the presented research results. Table 1 is the subject of a detailed analysis in this work. This analysis is unfounded. The operating parameters of the Varia R manually controlled fireplace should not be the values to which the operating parameters of a fireplace of another design have been compared.
9. In the work, descriptions under drawings should be brief and unambiguous. Designations on drawings should be unambiguously described under the drawing (a - description, b - description, c - description). Detailed descriptions of drawings should be in the text. Note applies to all drawings.
10. The description under Figure 7 should clearly indicate the conditions of each experiment. For example: what were the conditions of experiments 1, 2 and 3?, how did the conditions of experiment 4 and 5 differ?, etc.? If they were exactly the same, then this should be written. For the reader, the description of individual experiments 1 to 13 should be unambiguous, short and understandable, without looking for answers elsewhere in the paper.
11. On the left side of the graph in Figure 7, on the vertical axis is ,,Aver. CO emission / mg.min-1", and on the right, on the vertical axis is: Aver. CO emission / %. What is: "Aver. CO emission / %". This should be written unambiguously. The vertical axis of the graph, to the right, should be described differently.
12. The paper states that the tests were also carried out for pine wood. The paper should present the test results obtained during the combustion of pine wood and possibly compare them with the results obtained during the combustion of beech wood.
13. In the description of Figure 8, individual experiments should be clearly described. Differences between individual experiments should be clearly indicated.
14. In Figure 8, on the vertical axis on the left is the description: "Particle emission / mg.Nm-3". On the second vertical axis, on the right, is the description: "Particle emission / %". What does the description "Particle emission / %" mean? In my opinion, this description should be different.
15. The presentation of the research results is quite chaotic and makes the content difficult to understand.
Reviewer 2 Report
MS: sensors-2331536
Title: Extensive Emissions Reduction of Firewood-fueled Low Power Fireplaces: Advanced Combustion Airstream Control System with Long-Term Stable CO/HC Sensor and Oxidation Catalyst under real-life operation conditions
Using a sophisticated combustion air management system, a long-term stable CO/HC sensor, and an oxidation catalyst, this study proposes a novel and practical approach to significantly reducing emissions from low-power fireplaces fueled by firewood. After four months of testing in the field, the authors were able to prove that their technology reduced gaseous emissions by around 90% compared to conventional fireplaces that do not use a catalyst. This study explains why high-tech sensing equipment and feedback control loops are needed for effective autonomous regulation of combustion air streams in low-power fireplaces. This article can be included in “sensors” after minor revisions.
1. There is a dearth of a well-organized literature study regarding automated regulation of air flows for burning in low-power fireplaces.
2. Nevertheless, the authors do not include a thorough review or ablation experiments for the suggested technique, which would have shed more light on the efficacy and feasibility of their approach, if supplied.
3. The originality and importance of their proposed work would have been better highlighted if the study had included a comparison and discussion with well-established baselines in the area.
4. This study is incorrect in writing format. Please check the format at “sensors”.
5. The describe in Figure 1 is not suitable. The resolution of most Figures are too low to print for publisher. Some Figures should include the reference.
6. To make Tables more understandable, I advise the authors to rebuild them.
7. It is recommended to confirm the results of the simulation to ensure the accuracy of the results.
8. A few grammatical and typo mistakes are still there in your manuscript. Please thoroughly check and revise the manuscript carefully.
9. All units should be adopted according to consistent units. Literature writing format of this article is confused.
10. The content is plentiful, but some part of the reference literatures is kind of obsolete (in 5 years). Key publications should be cited as completed as possible. Please also clarify the novelty and application implication of your work in this section. I suggest authors refer to the latest literatures from “Sensors”, “MDPI”, and other disaster risk journals. But please do not exceed 25% of all citations from “Sensors”. Authors may see the following reference while revising. https://doi.org/10.1016/j.psep.2022.06.046ï¼›https://doi.org/10.1016/j.psep.2022.11.035
Reviewer 3 Report
Dear Editor and Authors,
I thank you for the opportunity to review this manuscript.
The article "Extensive Emissions Reduction of Firewood-fueled Low Power 2 Fireplaces: Advanced Combustion Airstream Control System 3 with Long-Term Stable CO/HC Sensor and Oxidation Catalyst 4 under real-life operation conditions" is presented in the required form. The topic studied in the article is interesting in several scientific areas from the point of view of energy, environmental pollution as parts of sustainable development.
This article needs some corrections and several additions to the following questions.
I think the title is too expanded, even 28 words, I recommend reducing it to no more than 20.
Introduction. The authors mention in the Introduction that "the use of biomass, especially of wood residues for residential heating, should be further promoted because heat from biomass combustion is well accepted as a renewable energy source. This is undoubtedly the main goal of sustainable development and the circular economy. I recommend providing citations of such studies, which may include:
https://doi.org/10.1016/j.jclepro.2017.10.241
https://doi.org/10.3390/en14248471
https://doi.org/10.1155/2019/7142804
The text is quite stretched, so it is difficult to determine the main idea of this study. It would have been logical for the authors to give one paragraph which would have been accurate and concise about the novelty of this study and its practical value for potential users of small boilers, i.e., the standard household, presumably a private home.
In this paper a great deal of attention is paid to CO/HC and oxygen sensors. However, the main research process to be monitored is the combustion process and pollutant emissions. I would like to understand why the authors avoid analyzing particulate matter? In the combustion of biofuels, only the particulate matter is of great importance, the authors have given only one Figure 8. It is due to the particulate emissions that biofuels are generally considered much more polluting than natural gas. Carbon monoxide is found in the combustion of both fuel options. Natural gas, of course, is not a renewable fuel in this case, a clear disadvantage.
As mentioned by the authors in the Introduction, the main biofuel for combustion should be from wood residues. In this study, quality firewood is used. According to the steps in the hierarchy, combustion is positioned as the last step in the treatment of both waste and energy production from raw materials. How did the authors only consider this research methodology?
Methodology. Why was this sequence of fuel loading and amount chosen for the experiments, e.g., cold loading of 5.5 kg and then three loads of 3.5 to 2.5 kg?
No information was found in the methodology on the solids measurement instruments/systems. How were the particulate matter reduction data obtained (Figure 8)?
It would be informative to indicate the main characteristics of the fuel (firewood) used in each phase of the study. Was there a significant/significant difference in the quality of firewood at different loads?
It would be informative to provide more detailed information on the parameters of the single room fireplace HKD7.
Reviewer 4 Report
In this paper, five different control algorithms are used to achieve combustion airflow control of log charge combustion, in order to accurately describe various combustion situations. Based on combustion experiments and on-site testing, it has been proven that using this long-term stable and advanced automatic combustion system can reduce the gas emissions associated with manually operated fireplaces without catalysts by about 90%. This study has important significance for reducing emissions only in household heating. It is recommended that the author make modifications according to the following comments.
1. In Abstract, it is recommended to provide the flue gas quality monitoring data to increase persuasiveness.
2. There are too many paragraphs in the Introduction, and there are too many redundant statements about PM emissions. It is recommended to simplify them.
3. The three pictures in Figure 2 should be marked with (a), (b), and (c) to make the reader more clear.
4. Section 4.2 (Validation of the control algorithm under real life operation conditions) should better confirm each other with the previous text, and it is recommended to supplement relevant descriptions.
5. In section 4.4 (Long term stability of the LH sensors), the results of this article should be compared with those of others to verify the long-term stability of the sensors in this study.
Round 2
Reviewer 3 Report
Dear Editor and Authors,
I thank you for the opportunity to 2nd review this manuscript.
The quality of article improved. Several additional questions/comments were presented.
If reducing particulate emissions is not the topic of this paper, I suggest adding the phrase "reducing gaseous emissions" to avoid confusing the readers.
In the experimental study, of course, simplifications are applied, so for this study, this methodology is appropriate. However, the question of the effectiveness of such emission reductions in real-world conditions using wood waste remains as a discussion, unlikely from a practical point of view, or requires fundamental improvements.
The manufacturer's advice is not a scientific approach to research. At this point, there are two solutions: 1) find out and describe why manufacturers recommend this type of loading; 2) by doing research, test different types of loading, which would be a valuable scientific result.
For this one-time study, this is accurate. However, for a long-term or repeat study, it will not be possible to properly compare the results due to insufficient information about the input data.
All true, but this study is presented by the authors and is not an advertising publication of the firm. Especially in the future, the homepage will be updated, and this information will be lost.
Reviewer 4 Report
I think it can be published
